# Exploring the BiFeO_3_-PbTiO_3_-SrTiO_3_ Ternary System to Obtain Good Piezoelectrical Properties at Low and High Temperatures

**DOI:** 10.3390/ma16216840

**Published:** 2023-10-24

**Authors:** Anton Tuluk, Sybrand van der Zwaag

**Affiliations:** Faculty of Aerospace Engineering, Delft University of Technology, Kluyverweg 1, 2629 HS Delft, The Netherlands

**Keywords:** bismuth ferrite, piezoelectric, ferroelectric, lead titanate, strontium titanate

## Abstract

In this work, we investigated the piezoelectric properties of BiFeO_3_-rich (1 − (y + x)) BiFeO_3_–y PbTiO_3_–x SrTiO_3_ (0.1 ≤ x ≤ 0.35; 0.1 ≤ y ≤ 0.3) bulk piezoceramics, as this system could potentially lead to the development of bulk piezoelectric ceramics that are suitable for high-temperature applications (>200 °C). Samples with various levels of PbTiO_3_ and SrTiO_3_ were prepared via a conventional solid-state route. X-ray diffraction confirmed a pure perovskite phase for the compositions, which was explored without secondary phases. It was found that the addition of comparable levels of PbTiO_3_ and SrTiO_3_ to the BiFeO_3_ ceramic resulted in higher piezoelectric properties compared to those of the pure BiFeO_3_ and binary systems. The Curie temperature was significantly reduced by dual doping, with SrTiO_3_ and PbTiO_3_ additions resulting in comparable Curie temperature depressions. The locations of the phase boundaries between the cubic, pseudocubic, and rhombohedral crystal structures were determined. The highest piezoelectric properties, including a d_33_ value of 250 pC/N at room temperature, were obtained for the samples with the composition x = 0.3, y = 0.25, which was close to the cubic–pseudocubic phase boundary in the phase diagram. The temperature dependence of the piezoelectric properties varied depending on the previous thermal history, yet an appropriate heat treatment resulted in an almost temperature-stable d_33_ value. The ceramic with the lowest temperature sensitivity and a high Curie temperature of 350 °C was found for x = 0.1, y = 0.2 with a d_33_ value of 60 pC/N at RT and 71 pC/N at 300 °C (after poling at 60 kV/cm and a stabilizing heat treatment). However, the materials developed were still unsuitable for applications at high temperatures due to a rapidly increasing electrical conductivity with increasing temperature.

## 1. Introduction

Piezoelectric sensors have long been the most efficient devices for generating and measuring dynamic forces, below or slightly above room temperature. For example, conventional ultrasonic echography measurements of the wall thickness of pipes and tanks offer advantages over other non-destructive testing methods, such as low cost, ease of operation, and the ability to conduct in operando measurements [1]. Piezoelectric sensors are already well established for this type of application but are limited to operating temperatures of up to 260 °C [2] due to most devices relying on the ubiquitous piezoelectric ceramic lead zirconate titanate or PZT. Therefore, considerable efforts are now aimed at developing materials that can satisfy the growing demand for piezoelectric elements with operating temperatures within the range of 350–600 °C.

One such material is BiFeO_3_ (BFO), which has an exceptionally high Curie temperature, TC, of 825 °C [3] and the additional benefit of being lead-free in composition. This high TC makes BFO an ideal candidate for use as a high-temperature lead-free piezoceramic [1]. Unfortunately, research on bulk BFO ceramics has not been as successful, as it has not demonstrated high piezoelectric coefficients and has had serious temperature issues. This is largely due to its high electrical conductivity at temperatures >200 °C, which limits the use of this material in real devices.

For high-temperature applications in actuators, transducers, sensors, etc., solid-state solutions of BiMeO_3_-PbTiO_3_ with a perovskite structure have been proposed as more sensitive prospective compositions due to their high Curie temperatures (TCs), especially those systems near morphotropic phase boundaries (MPBs). For example, the BiScO_3_-PbTiO_3_ (BS-PT) system was once expected to be a substitute for PZT for high-temperature applications due to its excellent performance near its MPB. However, the high cost of the raw material Sc_2_O_3_ made it unsuitable for commercial applications [4]. Also widely studied is the BiFeO_3_–PbTiO_3_ (BFO-PT) system, which has the highest TC (TC at MPB, approximately 630 °C) in the family of binary piezoceramics based on PT. However, it is difficult to obtain mechanically strong BFO-PT ceramics due to their negative temperature coefficient of expansion and high tetragonality, as in the case of pure PT ceramics [5]. In addition, the large tetragonal distortion (c/a ratio 1.18) for BFO-PT solid solutions results in a large coercive field of Ec > 100 kV/cm, making it difficult to have them fully poled [6].

It has been reported that a reduction in Ec and c/a of BFO-PT ceramics could be achieved through the substitution of large cations [7] to weaken the hybridization [8] between Pb^2+^/Bi^3+^ and O^2–^ ions as well as to strengthen the ferroelectric activity of B-site cations.

Popular strategies for improving piezoelectric responses reported in the literature include chemical modification (doping of elements, substitution or formation of new solid solutions) to reduce tetragonality, which can be achieved by replacing large cations to expand the a-axis and weakly ferroactive cations to shorten the c-axis [9,10]. For example, the introduction of La, Ca, Ba, or Sr into the A site, Ga, Al, Ta, or Nb into the B site, or a combination of PbZrO_3_, BaZrO_3_, or BaTiO_3_ with BFO-PT will lead to a significant decrease in the c/a ratio and a significant improvement in piezoelectric properties and in particular the mechanical stability. Most recently, BFO-PT-based ternary solid solutions, such as BFO-PT-BT, BFO-PT-BZ, and BFO-PT-BZT, have attracted great attention [11,12,13]. Preliminary experiments have shown high piezoelectric properties up to 300–400 pC/N (at room temperature), with a Curie temperature of approximately 300–400 °C. However, these piezoceramics generally have a high coercive field, up to 60–100 kV/cm, which creates difficulties in fully activating the piezoelectric properties upon poling.

The current study adopted the method of site engineering, introducing SrTiO_3_ (ST) into the BFO-PT system to form a ternary system. Despite the fact that ST itself does not possess piezoelectric properties at room temperature, it can still stabilize BFO-PT structures and decrease conductivity at RT and elevated temperatures. Furthermore, solutions of BFO-ST have been reported to have different structures at varying BiFeO_3_ contents, e.g., rhombohedral structures between 100 and 65 mol%, pseudocubic structures between 67 and 20 mol%, and cubic structures below 20 mol% of BFO [14]. The introduction of Sr^2+^ in the A site greatly influences the Pb/Bi–O bonding interaction and thus provides additional flexibility to tailor the dielectric, piezoelectric, and mechanical properties of BFO-PT-based solid solutions.

It is well known that BFO-PT and BFO-ST both can form solid solutions near the morphotropic phase boundary (MPB) region [14,15,16,17]. It is widely known that all functional properties of piezo materials with compositions near the MPB region increase due to the coexistence of two phases. For BFO-PT, the two phases in equilibrium are tetragonal and rhombohedral, and for BFO-ST, they are rhombohedral and cubic. For BFO-PT, a continuous solid solution across its entire composition range has been reported, with a rhombohedral–tetragonal MPB at 0.7–0.8 BFO.

The formation of ternary solid solutions provides additional degrees of freedom in the material design to improve the comprehensive properties and to meet various materials requirements.

In the present study, bulk ceramics with pseudoternary compositions of (1 − (y + x)) BFO–y·PT–x·ST (0.1 ≤ x≤ 0.35; 0.1 ≤ y ≤ 0.3) were explored to obtain a combination of a high T_C_ and good piezo properties at elevated temperatures. The ceramics were prepared by a conventional powder sintering method. The objectives of this study were: (1) to obtain high-strength piezoceramics with a low electrical conductivity at room temperature and a suitable coercive field for complete polarization, (2) to study the softening effect of ST on BFO-PT ceramics and determine the position of the phase boundary, and (3) to study the temperature dependence of the piezoelectric and electrical properties for the most promising systems.

## 2. Experimental Procedure

BiFeO_3_-PbTiO_3_-SrTiO_3_ (BFO-PT-ST) samples were prepared by a conventional solid-state reaction process using analytical-grade powders as raw materials: Bi_2_O_3_, Fe_2_O_3_, PbO, TiO_2_, and SrCO_3_. All powders were individually ground before weighing and mixing. Grinding and mixing were performed using yttria-stabilized ZrO_2_ balls in an isopropanol medium in polypropylene containers. The powders were subsequently dried and calcined at 775 °C for 1 h with a heating rate of 600 °C/h. After a slow cooling process, the calcined powders were reground, granulated by mixing with 2 wt% QPAC 40 binder, and then uniaxially pressed into disks (13 mm in diameter and approximately 1 mm in thickness) using a compaction pressure of 200–250 MPa. The final synthesis and sintering were performed at 1025, 1050, or 1075 °C for 1 h, depending on the system’s composition. Sintering parameters were optimized to achieve the highest density. In total, 15 different compositions were explored.

Scanning electron microscope (SEM) images were taken using a Jeol JSM 7500F field emission scanning electron microscope, Tokyo, Japan. Prior to SEM measurements, a thin (15 nm) layer of gold was deposited on the sample. To determine the phase purity, X-ray diffraction studies with Cu Kα radiation were conducted using a Rigaku MiniFlex tabletop XRD analyzer, Tokyo, Japan. X-ray diffraction studies with Cu Kα radiation were also performed using a Bruker D8 Discover diffractometer at elevated temperatures, Billerica, MA, USA. Scans were recorded at set temperatures: RT, 200 to 700 °C in 25 °C increments for selected systems. The density was determined by Archimedes’ method in an aqueous medium.

For electrical characterization, the samples were polished, coated with silver paste (DuPont 5220), and fired at 150 °C for 20 min. Polarization–electric field hysteresis loops were measured at room temperature using a Radiant precision ferroelectric analyzer. The samples were poled under a DC electric field of 20–80 kV/cm for 20 min in silicone oil at room temperature. After aging for 24 h, the piezoelectric constant d_33_ was measured with a quasi-static piezoelectric d_33_ meter. The Curie temperature was determined indirectly by determining the depolarization temperature. The electrical properties of samples at room temperature were measured using an Agilent 4263B LCR meter at 1 kHz and 1 V, Santa Clara, CA, USA.

Measurements of the high-temperature piezoelectric constant were carried out by a direct method using a laboratory setup from Kistler Instrumente AG at their lab in Winterthur. To this aim, samples were clamped between two Al_2_O_3_ disks with static preload of 1 MPa (100 N). To measure the piezoelectric response, a dynamic load of 0.1 MPa (10 N) with a frequency of 1, 2, 5, 10, 20, 50, 100 Hz was applied. Such measurements were performed at 22, 50, 100, 150, 200, 250, 300, 350 °C. When changing the temperature either during heating or cooling, the preload was removed. Up to 5 temperature cycles up to the maximum temperature have been performed to check the stability of the temperature dependency of the properties.

## 3. Results

### 3.1. Microstructure and Crystal Structure

Figure 1 shows two illustrative SEM images of a fresh fracture surface for the 0.6BFO-0.2PT-0.3ST ceramic. The surface features of the broken granules reveal equiaxed grains and randomly oriented domains. It is evident that these grains exhibit rather large dimensions, exceeding 100 nm. The grain size was determined by measuring a substantial number of grains in SEM images. The microstructure of ceramics does not depend much on the composition as shown in Appendix A. All binary and ternary ceramics produced in this work exhibit an average grain size of about 4–7 μm. An earlier study on BFO-PT ceramics reported a grain size of about 1 µm [1], while single-phase BFO sintered to a fully dense state could have a grain size of 10 µm [2]. The increase in the grain size in the ternary system with respect to the binary system might be due to a decrease in tetragonality upon the addition of ST. The relative density of the samples made was at least 97%. No significant correlation between sintered density and composition was identified.

Table 1 shows the unit cell parameters for all obtained compositions. The material crystallizes into a rhombohedral perovskite structure type with space group R3c at low SrTiO_3_ concentrations and undergoes a transition to a pseudocubic structure at higher concentrations of SrTiO_3_. All ceramics are found to exhibit a single-phase perovskite structure with no detectable secondary phases, indicating that the BFO-PT-ST formed a stable ternary perovskite solid solution. All systems could be classified into three possible types: cubic (C), pseudocubic (PC), and rhombohedral (Rh). We will consider a system to be of the rhombohedral type when the rhombohedral angle *γ*_rh_ is less than 89.95; of the pseudocubic type when *γ*_rh_ is close to 90 degrees but piezoelectric properties are still present; of the cubic type when *γ*_rh_ is 90 degrees and there are no piezoelectric properties. The obtained data clearly indicate that the addition of ST increases the symmetry of the unit cell.

It can be assumed that there should be a morphotropic phase boundary (MPB) or polymorphic phase boundary (PPB) [18] region in the BFO-PT-ST ternary system. It is located at the edge of the compositional domain where we have detected the pseudocubic crystal structure (0.45BFO-0.20PT-0.35ST; 0.50BFO-0.15PT-0.35ST; 0.40BFO-0.25PT-0.35ST).

### 3.2. Piezo- and Ferroelectric Properties at Room Temperature

Figure 2 shows the dependence of d_33_ at room temperature on the composition. It can easily be seen that an increase in piezoelectric properties occurs upon transition from a pseudocubic to a cubic crystal structure at an ST concentration of about 0.3. With a further increase in the ST concentration, although the transition from the pseudocubic to cubic phase has not occured, the piezoelectric properties start to diminish.

Figure 3 shows the variation of the Curie temperature with compositions in the BFO-PT-ST system. From Figure 3, it is clear that the Curie temperature decreases with decreasing BFO level. The range of Curie temperatures measured for these ternary systems is rather wide compared to those of modified PZT systems [9]. Near the phase boundary, the composition exhibits a Curie temperature as low as 30 °C, but as the concentration of ST decreases, it rises quite rapidly up to 600 °C at 0.1 mol ST. The difference in the contour plot between Figure 2 and Figure 3 (especially for high concentrations of BFO) can be explained by taking into account that for some compositions it was not possible to measure the piezoelectric constant, due to difficulties in prior polarization, as some BFO-based piezoceramics had a high coercive field, which made proper poling impossible. It should be pointed out that piezoelectric properties in samples depend not only on the crystallographic structure and composition but also on the polarizability of the material. A proper analysis of compositional effects requires that proper poling has been achieved in all samples. In the present work, all poling was carried out at room temperature with a maximum field strength of 50 kV/mm. The poling aspect will be further discussed in the Section 4.

Figure 4 shows the correlation between the piezoelectric constant at room temperature and the Curie temperature for the explored ternary compositions. As can be seen from Figure 4, there is a clear and seemingly unique correlation between the Curie temperature and the piezoelectric properties over the composition ranges explored. To put these values into perspective, the figure also shows the piezoelectric charge constants of commercial high-temperature piezoceramics: lead-based and lead-free. Although lead-based piezoceramics have high piezoelectric properties, they are limited to about 200–300 °C, in contrast lead-free ceramics allow this limitation to be exceeded by sacrificing high piezoelectric properties. For some concentrations, the ternary systems obtained in this study can compete with commercial materials.

Figure 5 presents the electrical resistivity values at room temperature for the compositions studied in the BFO-PT-ST system. At room temperature, all samples have a high resistivity exceeding 2 × 10^9^ Ohm·cm. The variations between the various samples show no obvious correspondence with crystal structure and may be due to slight variations in sintering temperature, precursor stoichiometry, presence of impurities, or moisture that occurred during laboratory sample preparation.

It has been observed that the piezoelectric properties depend not only on the composition and, coupled to this, the Curie temperature but also on the poling ability of the samples. Hence, it is important to determine the coercive field for the newly synthesized ceramics. Figure 5 and Figure 6 show P-E hysteresis loops measured at 5 Hz at room temperature for ceramics with varying concentrations of ST or PT. The difference in the resulting changes in the ferroelectric properties is obvious. With an increase in the ST concentration at a fixed PT concentration, the remnant polarization does not change, while the coercive field strongly decreases with an increase in the ST concentration, till it disappears at ST = 0.4. In contrast, the PT concentration not only affects the coercive field but also the remnant polarization and the shape of the hysteresis loop. The polarization value is gradually saturated with increasing electric field. A possible reason for this broad transition is the distribution of the locally required electric field to complete the switching mechanism, due to the coexistence of rhombohedral and tetragonal phases in the pseudocubic region. The coercive field in this case varies non-linearly with the change in the PT, but its behavior is consistent with that of the Curie temperature.

### 3.3. Performance at Elevated Temperatures

The impact of temperature on the piezoelectric charge constant (d_33_) values of three fully polarized ceramic (0.8 − x)BFO-0.2PT-xST (x = 0.1, 0.2, and 0.3), measured in situ, is shown in Figure 7. These compositions were chosen to study the temperature behavior of ceramics with significantly different Curie temperatures (T_c_ ≈ 600 °C for x = 0.1; T_c_ ≈ 450 °C for x = 0.2; T_c_ ≈ 350 °C for x = 0.3). For each material, the temperature dependence is measured in the pristine state and in three aged states. All measurements per composition were carried out successively on one sample each.

As the temperature increases, for the three studied compositions the piezoelectric response d_33_ also increases, with the temperature dependence being strongest for the highest ST concentration. It is worth noting that the temperature dependence of d_33_ decreases during subsequent measurement cycles. This may indicate some changes in the defect structure in the samples during annealing leading to stabilization of the piezoelectric properties. Interestingly, the d_33_ value continuously increases with temperature except for one case (Figure 7c, 2nd annealing run) where the d_33_ drops between 300 and 350 °C. This unique behavior will be explained in the Discussion.

Thermal stability was also determined by measuring d_33_ at a constant temperature of 300 °C as a function of the holding time. The results are shown in Figure 8. As can be seen, for all three compositions, d_33_ remains stable up to around 4 × 10^4^ s and then starts to decrease. The most stable ceramic is the one with the lower ST concentration, probably because it has the highest Curie temperature of 600 °C.

Finally, Figure 9 shows the temperature dependence of the electrical resistance of the three materials studied in more depth. At room temperature, all samples exhibit comparable high-resistance characteristics, which decrease rapidly as temperature increases. The temperature dependence did not seem to depend on composition and is comparable with that of pure BFO [4,5]. This observed low resistance at high temperatures restricts the suitability of this type of material for applications as the amplifying electronic circuitry cannot work effectively with such conductive materials. The high electrical losses originated from defect generation at elevated temperatures ultimately constrain the overall performance of the material. Therefore, the low resistance at high temperatures remains a critical issue that needs to be addressed in future studies.

To determine the impact of temperature on the defect dipoles and the internal bias field, we measured the S-E loops of the poled 0.6BFO-0.2PT-0.3ST ceramics at progressively higher temperatures and the results are shown in Figure 10 for temperatures up to 125 °C (holding time 15 min). This effect is reversible and after cooling to RT we obtained the asymmetric behavior again (cooling is slow, less than 5 °C/min). With increasing temperature, the shape of the S-E loops gradually changes from the initial asymmetric shape to the symmetric shape; the symmetric shape being the regular shape for poled ceramics. The above results indicate that the defect dipole effect weakens when increasing the temperature. The reason is mainly due to the oxygen vacancies having a large amount of energy to migrate at high temperature, resulting in the rearrangement of defect dipoles in the direction of the applied electric field. In this case, the defect dipoles of the poled ceramics decouple, and the internal bias field disappears, resulting in asymmetrical S-E loops. In future studies, it would be beneficial to explore this effect at higher temperatures and the reversibility at faster cooling rates. In the experiments of Figure 10, the temperature circuit was limited by the capability of the measuring setup.

## 4. Discussion

BFO-PT-ST was fabricated across the compositional space with the intention of replicating the MPB observed in PZT. The perovskite structure was formed across the entire compositional space with the absence of secondary phases. As indicated in Table 1, the BiFeO_3_-rich end of the phase diagram belonging to the rhombohedral phase was indexed to the R3c space group. The phase changed from pseudocubic to cubic as the concentration of BFO in the solution decreased. Similar to PZT [17], the piezoelectric properties significantly increase upon transitioning from the pseudocubic phase to the cubic phase, as opposed to the transition from the pseudocubic phase to the rhombohedral phase. As expected, this also reduces the giant tetragonality of 1.18 [5] and the electric coercive field of the system near the MPB, which hinders its use in applications. A possible explanation is an increase in poling efficiency due to a decrease in the coercive field with a decreasing amount of BFO in the system.

The ferroelectric Curie temperature was in wide range of 30–650 °C across the entire phase diagram and, as can be seen in Figure 3, the depolarization temperature was also largely commensurate with the Curie temperature. High-temperature stability was maintained for d_33_ until 20–50 °C below the TC. Figure 4 shows that all compositions fit a simple and unique relationship between the piezoelectric properties (at room temperature) and the Curie temperature very well. It should be noted that, according to the Landau theory [6], if the system experiences a second-order phase transition, all secondary parameters (dielectric permittivity, piezoelectric coefficient) increase a lot. This also explains the absence of a jump during the transition from the rhombohedral phase to the pseudocubic one, since this is a first-order phase transition.

Based upon this preliminary research into the BFO-PT system [1,16,17], it was hypothesized that the partial substitution of ST may reduce long-range non-centrosymmetric structural order and enhance piezoelectric performance. This could provide an explanation for the potential relationship between the increase in properties with similar ST and PT contents. Moreover, the influence of the relative concentrations of ST and PT on the crystal structure and the resultant piezoelectric properties merits further investigation.

The selected compositions for the d_33_ test at elevated temperatures demonstrated an increase in the magnitude of the d_33_ coefficient as the temperature increases, consistent with the Landau theory [6]. The d_33_ coefficient exhibited strong temperature dependence, as depicted in Figure 7 for compositions in the vicinity of the MPB. The increase in the piezoelectric charge coefficients d_33_ with increasing temperature is observed in many ferroelectrics including barium titanate and PZT [15,16]. Such an increase has also been reported in the binary BiScO_3_-PbTiO_3_ solid solutions [17]. The correlation between the d_33_ coefficient’s temperature behavior and the crystal structure is evident. Changes in the crystallographic phases with temperature modulate the piezoelectric properties, enhancing them close to phase transition temperatures.

In the ternary system with the highest ST content, a significant peak (i.e., the d_33_ coefficient increased by two times compared to room temperature) was observed in piezoelectric properties at 300 °C. This observation was attributed to a phase transition, as the Curie temperature for this sample was determined to be in the region of 350 °C.

Based on the obtained data, it can be concluded that the sample 0.7BFO-0.2PT-0.1ST can be utilized up to a temperature of 350 °C, 0.6BFO-0.2PT-0.2ST up to 150–200 °C, and 0.5BFO-0.2PT-0.3ST only at room temperature. At these maximum use temperatures, the d_33_ coefficient remains stable, with variations of no more than 10%. But, in contrast to PZT, the depolarization temperature is relatively high and approaches the phase transition temperature. The unusual enhancement of the d_33_ which we observe at 250–300 °C in the 0.5BFO-0.2PT-0.3ST ceramics can be practically useful. These findings provide important insights for the selection of appropriate compositions for piezoelectric applications under varying temperature conditions. However, the risk of depolarization in the vicinity of a phase transition can be a strong obstacle for practical applications at such high temperatures.

In practical applications, the repeatability of the material’s d_33_ during thermal cycling is maybe even more significant than the magnitude of its change. After several annealing cycles, as shown on Figure 7, not only did d_33_ at room temperature decrease but the temperature dependence also became weaker. As, for example, for the sample 0.6BFO-0.2PT-0.2ST: in the first annealing cycle, noticeable changes in d_33_ occur upon reaching a temperature of 150 °C, while during the second or third cycle this occurs already in the region of 200–250 °C. And, for the 0.5BFO-0.2PT-0.3ST sample: during annealing up to 300 °C, the d_33_ curves did not change, apart from after the 3rd cycle, when the temperature exceeded 350 °C (which is presumably higher than the Curie temperature). Hence, in the fourth cycle the piezoelectric response was significantly reduced. However, once this treatment was received the sample became more stable and showed no peaks in d_33_ at 300 °C. Similar results are seen when analyzing the bipolar strain curve; as the sample is heated, the stress asymmetry vanishes. This effect can presumably be explained by the domain pinning by oxygen defects similar to what occurs in hard-PZT ceramics [1,20,21]. This is when, at elevated temperatures, oxygen defects become more mobile and start concentrating on domain walls, thereby reducing the dipole moment (as a consequence of d_33_) but also preventing reorientation of domains (domain pinning) at lower temperatures, which leads to stabilization of d_33_. As shown in Figure 8, it is interesting that the decrease in piezoelectric properties at elevated temperatures does not occur immediately but after some time. It is possible that defects need time to diffuse to the domain walls and accumulate before a noticeable effect appears.

Although for all materials synthesized the electrical conductivity was high and adequate for use in sensors, the observed high electrical conductivity of the studied material at elevated temperatures, as shown in Figure 9, remains problematic. Varying the ST level had no noticeable effect on the temperature dependence of conductivity. Consequently, this makes the material unsuitable for high-temperature applications where stability and low electric conductivity are key factors. The high electrical conductivity may have originated from intrinsic defects or impurities present in the material. Further studies may be conducted to elucidate the origin of the high electrical conductivity and potentially mitigate this undesired effect in the material.

## 5. Conclusions

BiFeO_3_-rich BiFeO_3_-PbTiO_3_-SrTiO_3_ bulk piezoceramics with a perovskite structure can be made using a conventional solid-state process, achieving a density of over 97%. In perovskite BiFeO_3_ ceramic, PbTiO_3_ and SrTiO_3_ were well soluble at least until 35 mol%; no secondary phases were discovered. For samples close to the cubic–pseudocubic phase boundary in the phase diagram, the highest piezoelectric properties at room temperature are obtained. The Curie temperature is significantly reduced as a result of dual doping, and the relationship between it and BiFeO_3_ level is almost linear. Both SrTiO_3_ and PbTiO_3_ additions result in comparable Curie temperature depressions. For all compositions investigated, a unique relationship between T_C_ and d_33_ was found. Compared to pure BiFeO_3_ and binary systems, dual doping with PbTiO_3_ and SrTiO_3_ added at comparable levels results in significantly higher piezoelectric properties. It is discovered that the temperature dependence varies based on the previous thermal history. When an earlier heat treatment is carried out at a temperature higher than the used temperature, the temperature dependence decreases and stabilizes. For 0.6BiFeO_3_-0.2PbTiO_3_-0.2SrTiO_3_, the combination of a high T_C_ and an appropriate heat treatment result in a superior stable d_33_ value of 90 pC/N at room temperature (120 pC/N at 300 °C). However, the material remains unsuitable for applications at high temperatures due to the rapid increase in electrical conductivity with temperature. The SrTiO_3_ and PbTiO_3_ level seems to have no effect on the temperature dependency of conductivity. Future research should focus on these areas to gain a deeper understanding of the BiFeO_3_-PbTiO_3_-SrTiO_3_ ternary system and its potential applications.

## Figures and Tables

**Figure 1 materials-16-06840-f001:**
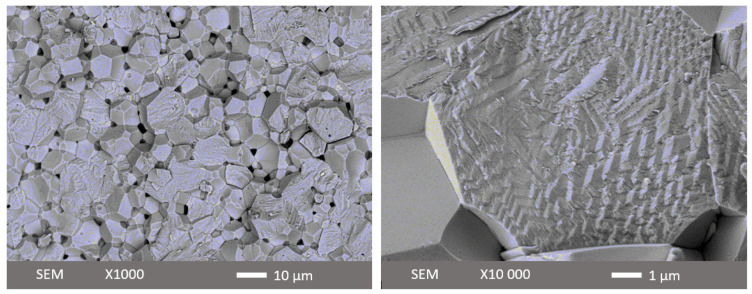
(SEM) images of a fractured 0.6BiFeO_3_-0.2PbTiO_3_-0.3SrTiO_3_ ceramic after sintering (acceleration voltage—5 kV).

**Figure 2 materials-16-06840-f002:**
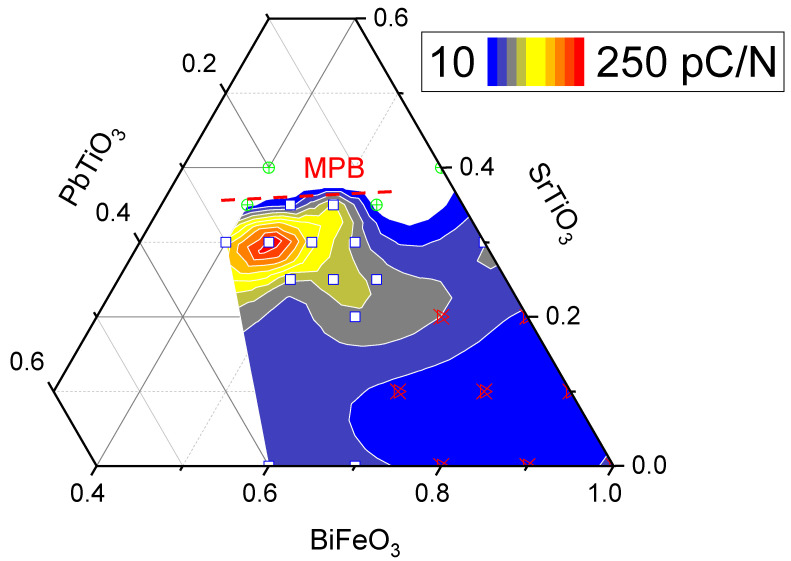
Values for the piezoelectric constant at room temperature for the ternary diagram of BFO-PT-ST (samples located in the white central region near BT = ST = 0.15 show no piezoelectric activity) (square—pseudocubic, PC; triangle—rhombohedral, Rh; circle—cubic, C; color scale is linear).

**Figure 3 materials-16-06840-f003:**
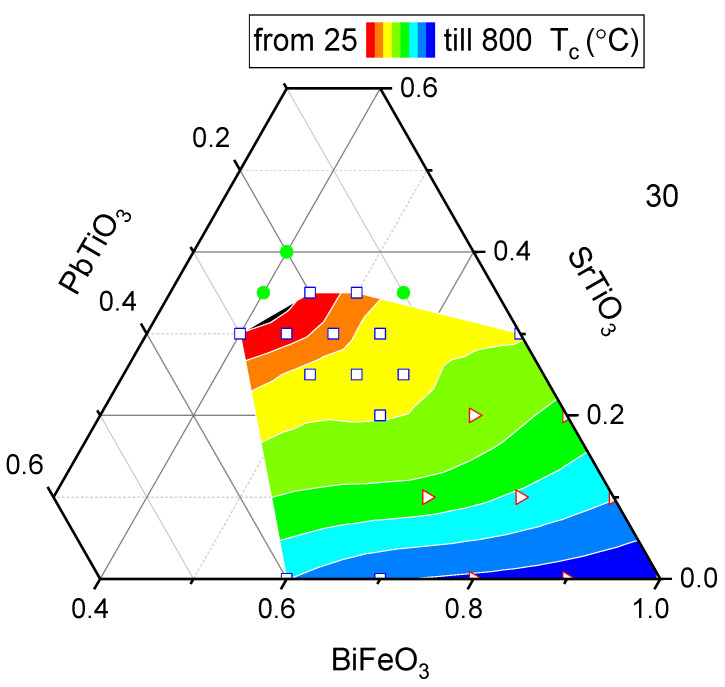
Isothermal map of Curie temperatures with various compositions in BFO-PT-ST ternary system (square—pseudocubic, PC; triangle—rhombohedral, Rh; circle—cubic, C; color scale is linear).

**Figure 4 materials-16-06840-f004:**
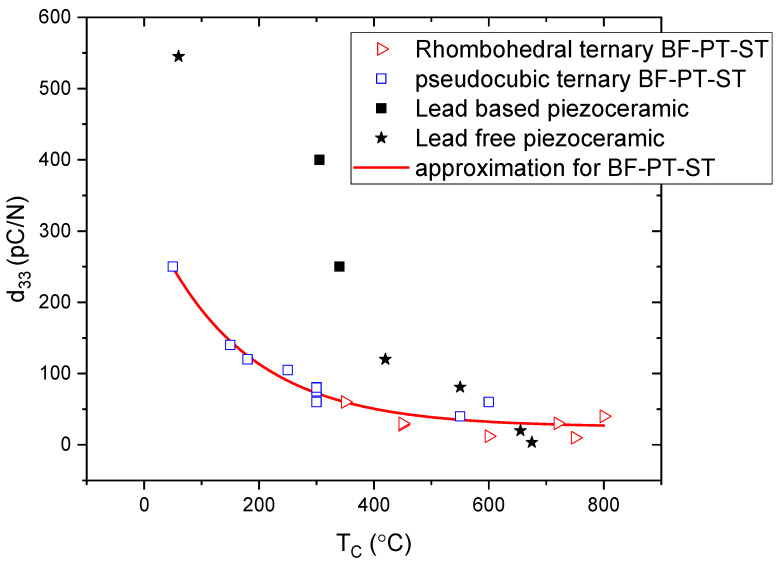
Relationship of piezoelectric properties at room temperature and Curie temperature [9,19].

**Figure 5 materials-16-06840-f005:**
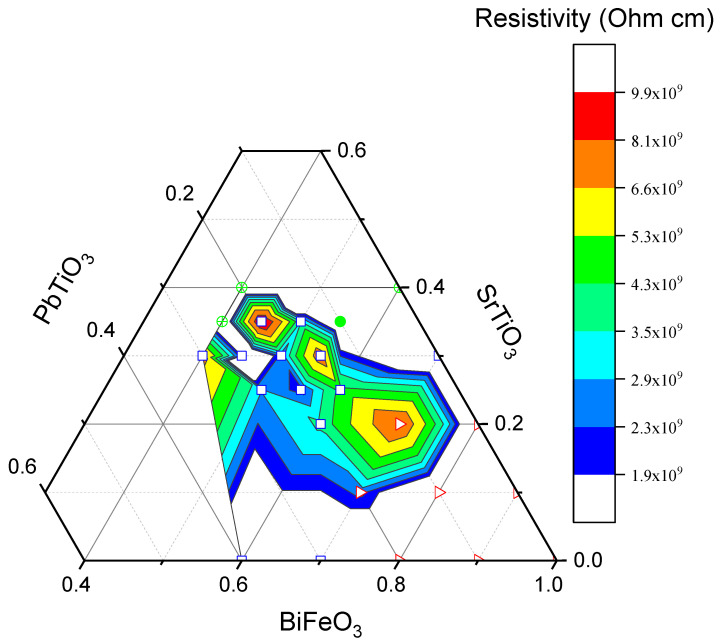
Electrical resistivity at room temperature for ternary diagram of BFO-PT-ST. The symbols correspond to the crystal structure as listed in Figure 3.

**Figure 6 materials-16-06840-f006:**
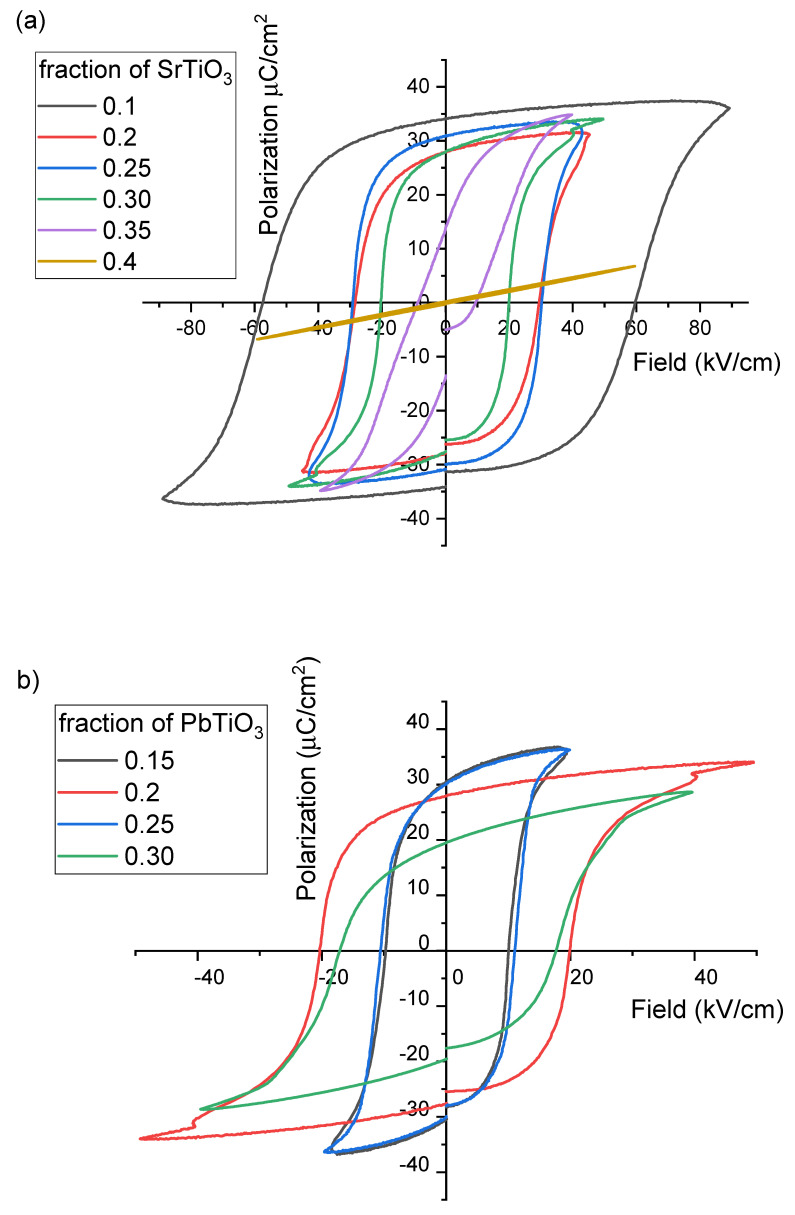
Polarization–electric field (P-E) hysteresis loop of the BFO-PT-ST sintered ceramics with (**a**) a fixed PT concentration (0.2PT) and (**b**) a fixed ST concentration (0.3ST).

**Figure 7 materials-16-06840-f007:**
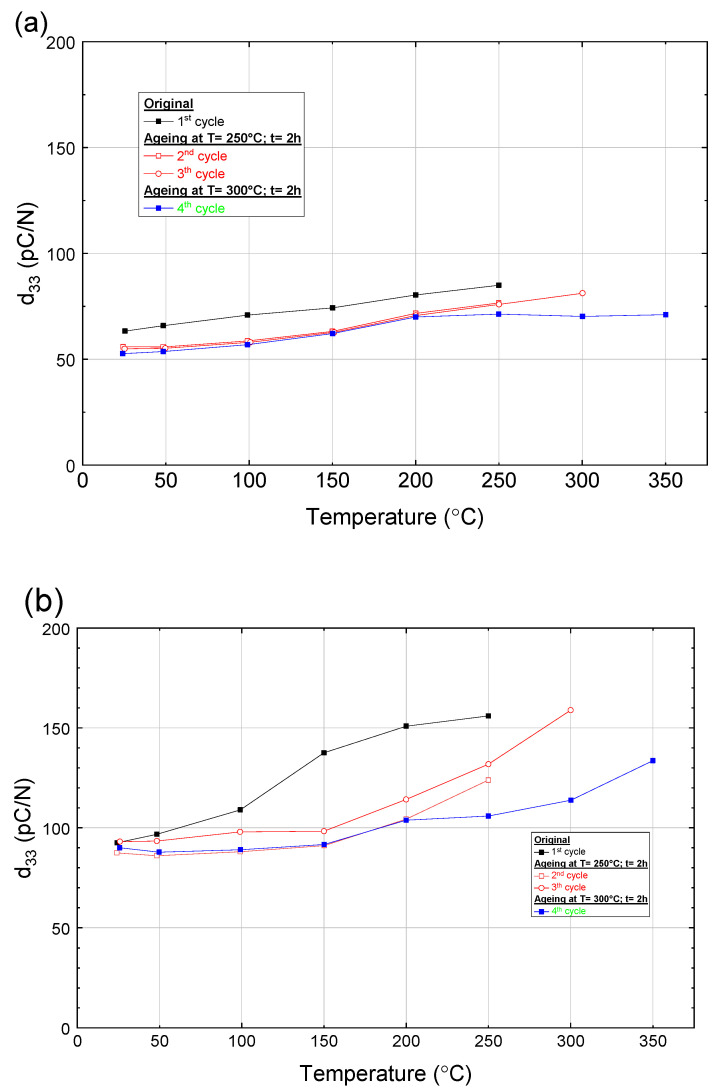
Temperature dependence of the piezoelectric properties of the (0.8 − x)BFO-0.2PT-xST ceramics for: (**a**) x = 0.1; (**b**) x = 0.2; (**c**) x = 0.3.

**Figure 8 materials-16-06840-f008:**
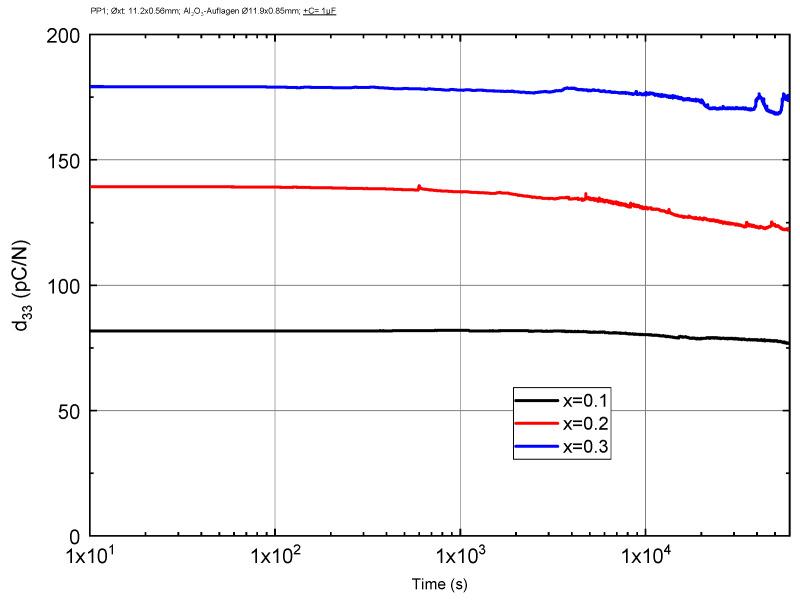
Dependence of piezoelectric constant d_33_ on holding time at 300 °C for (0.8 − x)BFO-0.2PT-xST ceramics.

**Figure 9 materials-16-06840-f009:**
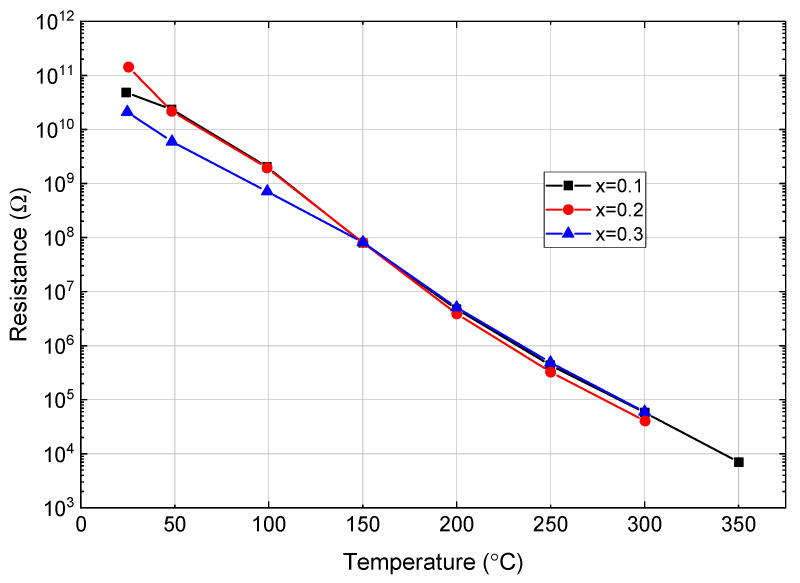
Temperature dependence of electrical resistance for (0.8 − x)BFO-0.2PT-xST ceramics.

**Figure 10 materials-16-06840-f010:**
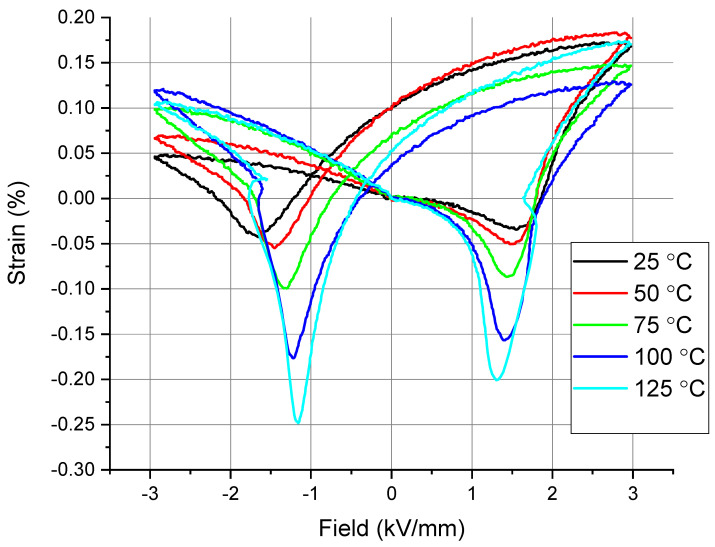
Bipolar strain curves of the 0.5BFO-0.2PT-0.3ST ceramic measured at different temperatures.

**Table 1 materials-16-06840-t001:** Unit cell parameter in hexagonal and tetragonal representation (C—cubic, PC—pseudocubic, Rh—rhombohedral, *a*_h_, *a*_rh_, *c*_h_, *γ*_rh_—unit cell parameters in h-hexagonal and rh-rhombohedral representation).

	Composition	Type	*a*_h_ (Å)	*c*_h_ (Å)	*a*_rh_ (Å)	*γ*_rh_ (deg)	d_33_ (pC/N)
I	0.40BFO-0.20PT-0.40ST	C	5.634	13.802	3.9846	90.001	0
	0.45BFO-0.20PT-0.35ST	PC	5.635	13.803	3.9846	89.998	25
	0.50BFO-0.20PT-0.30ST	PC	5.637	13.809	3.9861	89.995	140
	0.55BFO-0.20PT-0.25ST	PC	5.635	13.811	3.9862	89.991	105
	0.60BFO-0.20PT-0.20ST	PC	5.637	13.813	3.9864	89.984	81
	0.70BFO-0.20PT-0.10ST	Rh	5.631	13.861	3.9881	89.810	28
	0.80BFO-0.20PT-0.00ST	Rh	5.624	13.904	3.9891	89.645	30
II	0.70BFO-0.00PT-0.30ST	PC	5.597	13.727	3.9594	89.952	60
	0.55BFO-0.15PT-0.30ST	PC	5.635	13.811	3.9862	89.991	75
	0.50BFO-0.20PT-0.30ST	PC	5.637	13.809	3.9861	89.995	140
	0.45BFO-0.25PT-0.30ST	PC	5.636	13.804	3.9848	89.997	250
	0.40BFO-0.30PT-0.30ST	PC	5.641	13.808	3.9859	89.995	145
III	0.40BFO-0.25PT-0.35ST	C	5.639	13.811	3.9856	90.00	0
	0.50BFO-0.15PT-0.35ST	PC	5.635	13.811	3.9862	89.991	120
	0.55BFO-0.10PT-0.35ST	C	5.641	13.813	3.9863	90.00	0
	0.60BFO-0.15PT-0.25ST	PC	5.637	13.813	3.9864	89.984	77
	0.50BFO-0.25PT-0.25ST	PC	5.636	13.813	3.9867	89.989	80
	0.70BFO-0.10PT-0.20ST	Rh	5.635	13.829	3.9872	89.932	60
	0.70BFO-0.20PT-0.10ST	Rh	5.631	13.861	3.9881	89.810	28

## Data Availability

Data are contained within the article or Appendix A.

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
