# Peer review of "Exploring the BiFeO3-PbTiO3-SrTiO3 Ternary System to Obtain Good Piezoelectrical Properties at Low and High Temperatures"

_materials, 2023, doi:10.3390/ma16216840_

Round 1
Reviewer 1 Report
The authors present a study of the tuning of piezoelectric properties upon co-doping BiFeO3 with PbTiO3 and SrTiO3. The study, with a particular focus on high temperature piezoelectric performance, is well motivated, the experimental data of high quality, and the discussion fairly comprehensive and rigorous. The conclusions are convincing and fair, including stressing the remaining problem of too high conductivity at elevated temperatures, which will need to be addressed in future work. I recommend publication of the manuscript, after the authors have addressed the following points.
1. More references should be added, particularly in the introduction, and they should be specific. In many cases, there are statements for which no reference is given – it may be covered by an earlier or later reference on a more general statement, but it would be helpful for the reader to then repeat the reference where it specifically applies.
For just an example, for the paragraph starting with “Popular strategies for improving…” not a single reference is cited, and the paragraph contains several concrete statements including numbers.
2. The legend in Fig. 4 shows data with reference to [16] and [17]. There are only 15 references at the end of the paper.
3. The authors state that no correlation between sintered density and composition was observed. Nevertheless, it may be beneficial to include a table with the densities for each sample in the supporting information.
4. To facilitate linking structure and piezo-properties, adding a column to table 1 with the d33 at room temperature, would be helpful.
5. For the color scales in Figs. 2 and 3 it should be specified whether the scale is linear or logarithmic.
6. On p. 12, line 296, it is stated that “cooling is slow” – it would be nice to quantify this.
7. Some reading with regards to grammar and typo would be beneficial, in particular for the discussion section (e.g. polling instead of poling).
C.f. last point in report above. It is not that bad (somewhere between moderate and minor), but improvement possible
Author Response
Dear Reviewer,
We have taken all your comments into careful consideration and have made the necessary changes to address your concerns.
We have responded to each comment individually and have marked all corresponding changes in the article text in yellow for ease of reference. You will find our responses to each of points separately in attached file.

Reviewer 2 Report
This manuscript discusses about the system of BiFeO3-PbTiO3-SrTiO3 for being used as piezoceramics at low and high temperatures. Thus, it is suitable for being published in Materials.
However, some questions and concerns still need to be addressed:
Major:
1.) Page 1, line 9: could the authors comment on what the temperature range is defined as “high-temperature” here ?
2.) Page 1: The title emphasizes both low-temperature and high-temperature applications, however, the manuscript only talks about the importance of high-temperature, could the authors please comment on this ?
3.) Page 3, line 115 to 122: could the authors please provide information for SEM: which acceleration voltage and detector were used here
4.) Page 4, line 161: could the author comment on why the obtained particles are defined as a solid solution, is it possible to a composite formed ? Could the authors please show EDS mapping for the obtained particles ?
5.) Page 15: there is a big concern about the references cited here: 1.) there are 15 references in total for this manuscript, which are not enough for a thoughtful paper; 2.) among all these references, the latest ones are two from 2021 and one from 2020. Could the authors comment on why this field has so limited progress in recent years ?
Minor:
1.) Page 4, Table 1: could the authors add error bars to the lattice parameters ?
Based on the current status of this manuscript, I recommend a minor revision.
the English language quality is fine, just need minor editing
Author Response

(The authors gave the same response as above.)

Reviewer 3 Report
This manuscript investigated the piezoelectric properties and various electrical properties of the BF-PT-ST turnery system. Overall, this manuscript lacks scientific theory and some results. It is difficult to understand what the author is trying to argue. The authors require scientific discussion for this manuscript to be published. At last, the grammar and sentence structure should be significantly polished. Scientific approaches and explanations also need to be substantially refined. In short, this manuscript lacks inspiring insight. I would not recommend it for publication in Materials at this stage.
1. How to calculate the lattice constant and rhombohedral distortion? The author should provide the full XRD data in the manuscript. In addition, in order to understand the crystal structure, it is necessary to provide the manuscript with an enlarged drawing of some of the peaks as a (111) and (200).
2. How do you measure Curie temperature? The authors should provide the results of temperature-dependent dielectric constant measurement. Is there a dielectric relaxation seen in relaxor ferroelectrics?
3. Why is ferroelectricity exhibited in a pseudo-cubic structure? Authors should be explain for reason of this phenomenon.
The grammar and sentence structure should be significantly polished. I found a lot of typos in the manuscript. the structure of the sentence and contents are very difficult to understand. I highly recommend you to do English correction.
Author Response
Dear Reviewer,
We have taken all your comments into careful consideration and have made the necessary changes to address your concerns.
We have responded to each comment individually and have marked all corresponding changes in the article text in yellow for ease of reference. Below, you will find our responses to each of points separately in attached file.

Reviewer 4 Report
The paper entitled "Exploring the BiFeO3-PbTiO3-SrTiO3 ternary system to obtain good piezoelectrical properties at low and high temperatures" by Anton Tuluk and Sybrand Van der Zwaag presents the piezoelectric properties of BiFeO3-rich (1-(y+x))·BiFeO3– 7 y·PbTiO3–x·SrTiO3 (0.1 ≤ x ≤ 0.35; 0.1 ≤ y ≤ 0.3) bulk piezoceramics as this system could potentially 8 lead to bulk piezo electric ceramics suitable for high-temperature applications.
The paper needs major revisions, as follows:
1) The authors claim they preformed X-ray diffraction analysis (in abstract and at page 3, at the Experimental Procedure part), but it is no Figure in the paper presenting the diffractograms. Table 1 must be correlated with the X-ray diffraction metod results and also the database used for the crystalographic structure determination must be precised.
2) It can be made a correlation between crystal structure and the variation of thermal stability presented in figure 7?
Minor english corrections must be made
Author Response
Dear Reviewer,
We have taken all your comments into careful consideration and have made the necessary changes to address your concerns.
We have responded to each comment individually and have marked all corresponding changes in the article text in yellow for ease of reference. Below, you will find our responses to each of points in attached file.
Round 2
Reviewer 3 Report
Can be accepted
It is recommended to have a native speaker's proofreading before publication.
Reviewer 4 Report
The paper can be published in this form